# Transcriptome Analysis of Sake Yeast in Co-Culture with *kuratsuki Kocuria*

**Karin Kobayashi and Hiromi Nishida \***

Graduate School of Food and Nutritional Sciences, Toyo University, 48-1, Oka, Asaka 351-8510, Saitama, Japan; s3c102400021@toyo.jp
* Correspondence: nishida038@toyo.jp

**Abstract:** *Kuratsuki* bacteria enter the sake production process and affect the flavor and taste of sake. This study compared gene expression in the sake yeast *Saccharomyces cerevisiae* in co-culture with *kuratsuki Kocuria* to that in monoculture. Among the 5922 genes of *S. cerevisiae*, 71 genes were upregulated more than 2-fold, and 61 genes were downregulated less than 0.5-fold in co-culture with *kuratsuki Kocuria*. Among the stress-induced genes, fourteen were upregulated, and six were downregulated. Among the fourteen upregulated genes, six were induced in response to replication stress. Although the G1 cyclin gene *CLN3* was upregulated by more than 2-fold, eight genes that were induced in response to meiosis and/or sporulation were also upregulated. Fourteen metabolism-related genes, for example, the glyceraldehyde-3-phosphate dehydrogenase genes *TDH1*, *TDH2*, and *TDH3*, were downregulated by less than 0.5-fold in co-culture with *kuratsuki Kocuria*. The gene expression patterns of *S. cerevisiae* co-cultured with *kuratsuki Kocuria* differed from those co-cultured with lactic acid bacteria. Therefore, *S. cerevisiae* responded differently to different bacterial species. This strongly suggests that *kuratsuki* bacteria affect gene expression in sake yeast, thereby affecting the flavor and taste of sake.

**Keywords:** gene expression; *kuratsuki* bacteria; *Kocuria*; microbial interaction; sake yeast; *Saccharomyces cerevisiae*





## 1. Introduction

Sake is a traditional Japanese alcoholic beverage composed of *koji*, *moto*, rice, and water. *Koji* is produced by growing the *koji* mold *Aspergillus oryzae* on steamed rice. The enzymes of *A. oryzae* digest rice and produce sugars and amino acids; however, they cannot produce ethanol. *Moto* is a fermentation starter made from a mixture of *koji*, rice, water, and sake yeast *Saccharomyces cerevisiae*. *S. cerevisiae* converts sugar into ethanol, which is used in beer, sake, and wine production [1]. The flavor and taste of sake are mainly determined by the composition of esters and organic acids that are produced by the sake yeast [2–6]. Thus, different strains of sake yeast produce different flavors and determine the taste of the sake [7]. Some lactic acid bacteria interact with *S. cerevisiae* to change its metabolism, which results in the specific flavor and taste of the alcoholic beverage [8–10]. The effects of co-culture of *S. cerevisiae* with bacteria were reported in the industrial ethanol production process [11].

Bacteria that live in a sake brewery are called *kuratsuki* bacteria [12,13]; these bacteria inevitably enter the sake production process. *Kuratsuki* bacteria have evolved to enable adaptation to the sake-making environment through the modification of their genomes [13–15]. Although research on *kuratsuki* yeasts has been performed, research on *kuratsuki* bacteria has been considered unnecessary. As a result, there was no information on the difference in *kuratsuki* bacteria among different sake breweries. To elucidate the diversity of *kuratsuki* bacteria, we isolated and identified bacteria from the sake production process and subsequently characterized them [14,16].

Strains belonging to the genus *Kocuria* were isolated and identified as *kuratsuki* bacteria from the Narimasa Sake Brewery in Toyama, Japan [15,17]. *Kocuria* belongs to the phylum Actinobacteria and is not a lactic acid bacterium. We isolated and identified two different strains belonging to the genus *Kocuria*, which were *K. koreensis* strain TGY1120_3 and *K. uropygioeca* strain TGY1127_2 [15]. We have used these two strains as representative *kuratsuki* bacteria in our experiments, especially *K. uropygioeca* strain TGY1127_2 [18–20].

Although this strongly suggests that *kuratsuki* bacteria may interact with sake yeast to change the metabolism as well as lactic acid bacteria, which results in the taste of sake [14,18–21], the function of *kuratsuki* bacteria remains uncertain. We used the taste recognition device to confirm whether the co-culture would change the taste. Different effects of *kuratsuki* bacteria were found on different types of *koji* and different strains of sake yeast [13,18,19]. Thus, the taste of sake with *kuratsuki* bacteria differed from that without *kuratsuki* bacteria. We are performing research focusing on microbial interaction.

Comparison of gene expression is one of the most effective methods for elucidating the interactions between sake yeast and bacteria. There have been no reports on the genome-wide gene expression analysis of sake yeast in response to *kuratsuki* bacteria. In this study, we compared the gene expression of sake yeast in monoculture and co-culture with that of *kuratsuki Kocuria*.

## 2. Materials and Methods

### 2.1. Culture of Sake Yeast and kuratsuki Kocuria

The sake yeast *S. cerevisiae* strain K1401 [7] and *Kocuria uropygioeca* strain TGY1127_2 [15] were used in this study. Each strain was pre-cultivated in TGY medium (5 g/L tryptone, 1 g/L glucose, and 3 g/L yeast extract) at 25 °C for 12 h. One milliliter of pre-cultivated cell suspension of K1401 ($2.25 \times 10^6$ cells/mL) was added to 200 mL of modified TGY medium (20 g/L tryptone, 150 g/L glucose, and 10 g/L yeast extract). In addition, 1 mL of the pre-cultivated cell suspension of K1401 and 1 mL of the pre-cultivated cell suspension of TGY1127_2 ($>1 \times 10^8$ cells/mL) were mixed with 200 mL of modified TGY medium. Thus, monoculture and co-culture were performed in the artificial medium, which differs from the fermentation environment during sake making. The two solutions were incubated at 14 °C for seven days.

### 2.2. Measurement of Brix

Brix was measured daily (on days 0, 1, 2, 3, 4, 5, 6, and 7) using a PAL-BX/ACID digital refractometer (ATAGO, Tokyo, Japan). Brix is a measure of the percentage of soluble solids in syrups and products. Each degree of Brix is equal to 1% concentration of sugar in solution when measured at 20 °C (https://www.foodscience-avenue.com/ (accessed on 20 April 2024)). Each measurement was repeated thrice. The median of the three values was selected as the representative value. The Kolmogorov–Smirnov test was performed to compare the two samples using R software (The R Project for Statistical Computing, http://www.R-projet.org/ (accessed on 25 January 2024)). The Kolmogorov–Smirnov test is sensitive to differences in both the location and the shape of the empirical cumulative distribution functions of the two samples, which is one of the most useful nonparametric methods for the comparison of two samples.

### 2.3. RNA Isolation and Sequencing

DNA was digested and total RNA was isolated using an RNeasy Mini Kit (Qiagen, Hilden, Germany). After removing rRNA with riboPool (siTOOLs Biotech, Planegg, Germany), sequencing libraries were prepared using the MGIEasy RNA Directional Library Prep Set (MGI Tech, Shenzhen, China). The resulting libraries were analyzed using Synergy H1 (Agilent, Santa Clara, CA, USA) and QuantiFluor dsDNA System (Promega, Madison, WI, USA). The MGIEasy Circulation Kit (MGI Tech, Shenzhen, China), DNBSEQ DNB Rapid Make Reagent Kit (MGI Tech), and DNBSEQ-T7 DNB Rapid Load Reagent Kit (MGI Tech) were used. Sequencing was performed ($2 \times 150$ bp) using a DNBSEQ-T7RS

High-throughput Sequencing Kit (MGI Tech) and DNBSEQ-T7 (MGI Tech). The adapter sequence was removed using Cutadapt (ver. 4.0); bases with a quality score of less than 20 and paired reads with a length smaller than 75 bases were removed using Sickle (ver. 1.33). High-quality reads were mapped to the *Saccharomyces cerevisiae* S288C genome (NCBI RefSeq ID: GCF_000146045.2).

### 2.4. Gene Expression Comparison

We calculated the ratio of reads per kilobase of exon per million mapped reads (RPKM) [22] in sake yeast in co-culture with the *K. uropygioeca* TGY1127_2 per RPKM in the sake yeast in monoculture and selected a ratio of >2 for the upregulated genes and a ratio of <0.5 for the downregulated genes.

## 3. Results and Discussion

### 3.1. Brix of Culture

The Brix change pattern of the co-culture of sake yeast K1401 and *K. uropygioeca* TGY1127_2 was similar to that of the monoculture of sake yeast (Figure 1, Table S1). These patterns were not significantly ($p > 0.05$) different in the Kolmogorov–Smirnov test. This indicates that *kuratsuki Kocuria* did not inhibit alcohol fermentation by sake yeast, which is consistent with a previous study [18]. The 7-day monoculture and co-culture system was used for RNA isolation and sequencing. Brix was decreasing at 7 days (Figure 1, Table S1), thereby indicating that ethanol fermentation by sake yeast is in progress.

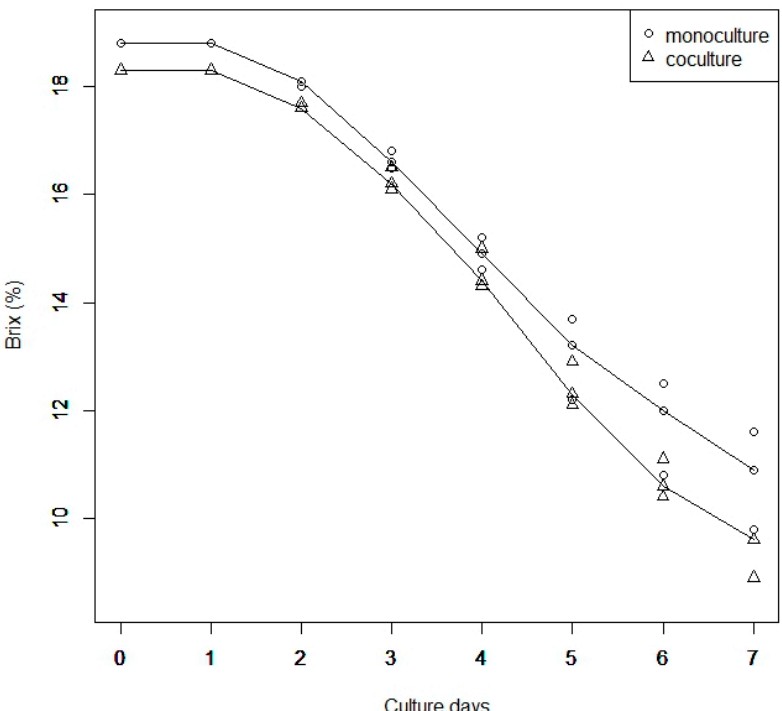

**Figure 1.** Brix of monoculture of the sake yeast K1401 and co-culture with the *kuratsuki Kocuria* TGY1127_2. These change patterns were not significantly different ($p > 0.05$ in the Kolmogorov–Smirnov test).

### 3.2. RNA Mapping and Gene Expression

The number of high-quality paired reads was 26,629,768 and 33,522,652 in the monoculture of sake yeast K1401 and co-culture with *K. uropygioeca* TGY1127_2, respectively. Among these reads, 24,367,924 (91.59%) and 30,433,279 (90.87%) reads were mapped to a single locus in the sake yeast monoculture and co-culture with *kuratsuki Kocuria*, respectively. In this study, we did not use any reads that were mapped to multiple loci because their origins were unclear, but we did use the reads that were mapped to a single locus.

We mapped the RNA sequences of 6420 genes of *Saccharomyces cerevisiae*. The number of genes that mapped more than 10 sequences in both sake yeasts in monoculture and co-culture with *kuratsuki Kocuria* was 5922. For these 5922 genes, the ratio of RPKM (the ratio of reads per kilobase of exon per million mapped reads) in co-culture with *kuratsuki Kocuria* per RPKM in monoculture. The mean ratio was 0.955 ± 0.008. We compared the gene expression levels of 5922 genes in sake yeast K1401 in monoculture and co-culture with *K. uropygioeca* TGY1127_2. Among the 5922 genes, 71 genes were upregulated more than 2-fold, and 61 genes were downregulated less than 0.5-fold in co-culture with *kuratsuki Kocuria* (Figures 2 and 3, Tables S2 and S3, Figure S1).

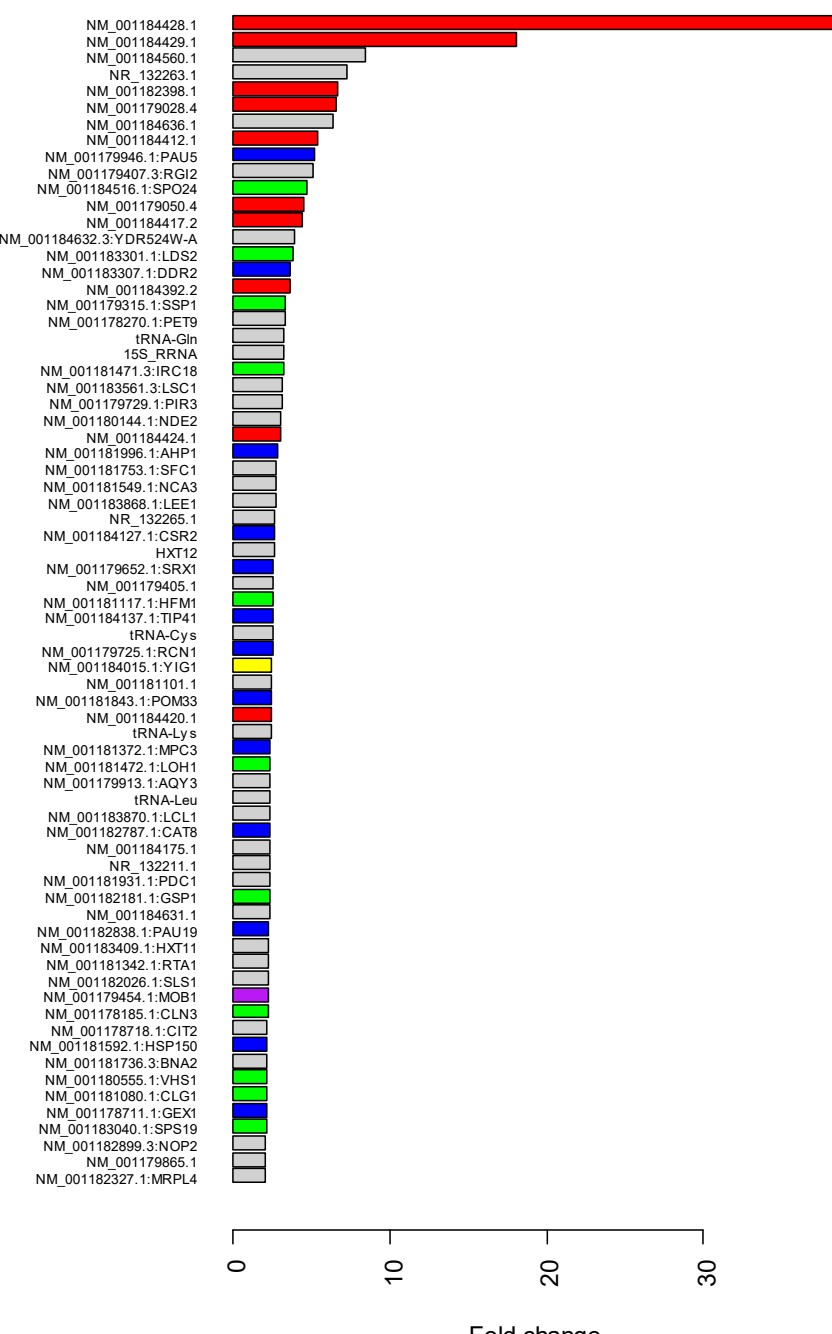

**Figure 2.** More than 2-fold upregulated genes of sake yeast in co-culture with the *kuratsuki Kocuria*. Blue, green, red, and yellow indicate the genes involved in stress response, cell cycle, and/or sporulation, retrotransposons, and metabolism, respectively. Purple indicates the gene involved in stress response and cell cycle. Grey indicates the other genes.

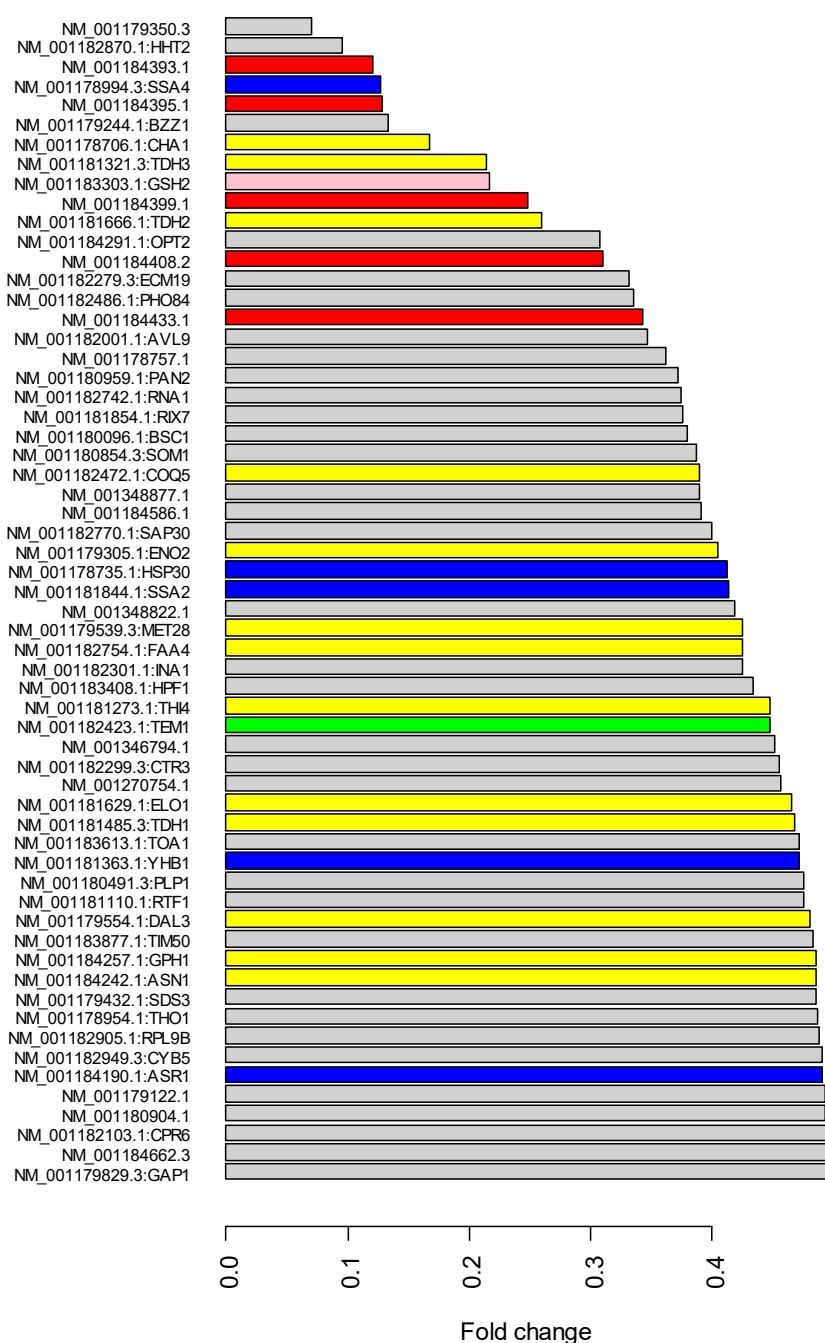

**Figure 3.** Less than 0.5-fold downregulated genes of sake yeast in co-culture with the *kuratsuki Kocuria*. Blue, green, red, and yellow indicate the genes involved in stress response, cell cycle, and/or sporulation, retrotransposons, and metabolism, respectively. Pink indicates the gene involved in stress response and metabolism. Grey indicates the other genes.

### 3.3. Gene Expression Changes of Retrotransposons and Small Nuclear RNAs in Co-Culture with the kuratsuki Kocuria

The most (38.8-fold) and second-most (18.0-fold) upregulated genes in co-culture with *K. uropygioeca* TGY1127_2 encoded a retrotransposon TYA gag protein (Figure 2, Table S2). Various stressors activate *S. cerevisiae* retrotransposons [23–27]. In our data, although 10 genes that encode the retrotransposon TYA gag protein were upregulated more than 2-fold in co-culture with *kuratsuki Kocuria*, five genes were downregulated less than 0.5-fold (Figures 2 and 3, Tables S2 and S3). This suggests that *K. uropygioeca* TGY1127_2 affects

retrotransposon regulation in sake yeast. Further studies are needed to elucidate yeast genome modifications using retrotransposons induced by *kuratsuki Kocuria*.

The small nuclear RNAs SNR8 and SNR5, were 7.3-fold and 2.6-fold, respectively, in co-culture with *K. uropygioeca* TGY1127_2 (Figure 2, Table S2). These two small nuclear RNAs are hydrogen bound to 20S pre-rRNA [28]. In this study, 81 snRNAs were identified. Interestingly, nuclear small RNAs other than SNR5 and SNR8 were not upregulated more than 2-fold (Figure 2, Table S2). In addition, *NOP2* was 2.1-fold upregulated in the co-culture with *kuratsuki Kocuria* (Figure 2, Table S2), which encodes a ribosomal RNA (cytosine-C5-) methyltransferase [29]. This suggests that sake yeast activates the processing of pre-ribosomal RNA when co-cultured with *kuratsuki Kocuria*.

### 3.4. Expression Changes of Stress-Related Genes in Co-Culture with the kuratsuki Kocuria

Among the 71 more than 2-fold upregulated genes in the co-culture with *K. uropygioeca* TGY1127_2, the following fourteen were stress-induced genes: *AHP1*, *CAT8*, *CSR2*, *DDR2*, *GEX1*, *HSP150*, *MOB1*, *MPC3*, *PAU5*, *PAU19*, *POM33*, *SRX1*, *TIP41*, and *RCN1* (Figure 2, Table S2). Ahp1 is involved in oxidative stress response [30]. The transcription factor Cat8 is involved in the reprogramming of carbon metabolism [31,32]. Csr2 is involved in osmotic stress response [33]. Ddr2 is involved in response to multiple stresses [34,35]. Gex1 is a glutathione exchanger that is induced in response to ion depletion [36]. Hsp150 is involved in the response to aluminum and oxidative stress [37,38]. Mob1 is involved in the regulation of mitotic exit [39,40]. Mpc3 stimulates respiration, which is involved in response to oxidative stress [41,42]. *S. cerevisiae* contains 24 Pau proteins [43]. Pau5 and Pau19 are structurally different [43]. Pau5 may be involved in the recombination of transposable elements [43,44]. The upregulation of *PAU5* may be related to retrotransposon activation. Pom33 is a membrane protein of the nuclear pore complex [45,46]. Srx1 is involved in the oxidative stress response [47]. Tip41 is involved in the negative regulation of the TOR signaling pathway [48]. Rcn1 is involved in the response to ethanol stress [49]. Among these fourteen stress-induced proteins, the abundance of six proteins, Cat8, Mob1, Mpc3, Pom33, Srx1, and Tip41, increased in response to DNA replication stress [50].

Replication stress inhibits transcription of histone genes [51]. In our data, seven of ten histone genes were downregulated in the co-culture with *kuratsuki Kocuria* (Table 1). Among the histone genes, *HHT2* was downregulated (Figure 3, Table 1 and S3). In addition, *RTF1*, *SAP30,* and *SDS3* were downregulated by less than 0.5-fold in co-cultures *K. uropygioeca* TGY1127_2 (Figure 3, Table S3) [52–54]. Therefore, *kuratsuki Kocuria* is strongly suggested to increase replication stress in sake yeast.

**Table 1.** Expression change of histone genes of sake yeast in co-culture with the *kuratsuki Kocuria*.

| Genbank ID | Gene | Product | Length (nt) | No. of Mapped Reads | | RPKM | | Ratio |
|---|---|---|---|---|---|---|---|---|
| | | | | Monoculture | Coculture | Mono | Co | Co/Mono |
| NM_001182870.1 | *HHT2* | histone H3 | 411 | 36894 | 4354 | 3749.46 | 358.96 | 0.096 |
| NM_001178358.1 | *HHT1* | histone H3 | 411 | 2419 | 1624 | 245.84 | 133.89 | 0.545 |
| NM_001178243.1 | *HTA2* | histone H2A | 399 | 2201 | 1517 | 230.41 | 128.83 | 0.559 |
| NM_001178357.1 | *HHF1* | histone H4 | 312 | 1616 | 1169 | 216.34 | 126.96 | 0.587 |
| NM_001180533.3 | *HTA1* | histone H2A | 399 | 4178 | 3769 | 437.37 | 320.08 | 0.732 |
| NM_001178242.1 | *HTB2* | histone H2B | 396 | 829 | 749 | 87.44 | 64.09 | 0.733 |
| NM_001182869.1 | *HHF2* | histone H4 | 312 | 3556 | 3677 | 476.06 | 399.34 | 0.839 |
| NM_001183941.1 | *HHO1* | histone H1 | 777 | 936 | 1194 | 50.32 | 52.07 | 1.035 |
| NM_001183266.1 | *HTZ1* | histone H2AZ | 405 | 445 | 588 | 45.89 | 49.20 | 1.072 |
| NM_001180532.3 | *HTB1* | histone H2B | 396 | 600 | 855 | 63.29 | 73.16 | 1.156 |

In contrast, six stress-induced genes, *ASR1*, *GSH2*, *HSP30*, *SSA2*, *SSA4*, and *YHB1*, were downregulated by less than 0.5-fold in co-culture with *kuratsuki Kocuria* (Figure 3, Table S3). Asr1 is involved in the response to ethanol stress [55]. Gsh2 is a glutathione synthase, which is involved in response to oxidative stress [56,57]. Hsp30 is involved in the re-

sponse to several stressors [58,59]. Ssa2 and Ssa4 are a member of Hsp70 chaperones [57,60]. Yhb1 is involved in the oxidative stress [61]. Thus, the expression of some stress-induced genes was diametrically opposed; for example, *ASR1* vs. *RCN1* in response to ethanol stress and *AHP1* vs. *YHB1* in response to oxidative stress. These results suggest that the stress levels of sake yeast co-cultured with *kuratsuki Kocuria* may not be as high.

### 3.5. Expression Changes of Cell Cycle-Related Genes in Co-Culture with the kuratsuki Kocuria

The following eight genes, *HFM1*, *IRC18*, *LDS2*, *LOH1*, *MOB1*, *SPO24*, *SPS19*, and *SSP1,* are involved in meiosis and/or sporulation, and were upregulated more than 2-fold in co-culture with *K. uropygioeca* TGY1127_2 (Figure 2, Table S2). Hfm1 is a DNA helicase involved in meiotic crossing over [62]. Irc18 and Lds2 are involved in spore wall assembly [63,64]. Loh1 is involved in the loss of heterozygosity [65]. As previously mentioned, stress-induced Mob1 is involved in the regulation of mitotic exit [40]. *SPO24* produces two mRNA forms, which appear during meiosis and mid-late meiosis [66]. Sps19 is a 2, 4-dienoyl-CoA reductase involved in sporulation [67]. Ssp1 is involved in meiotic division and spore formation [68].

In addition, four genes, *CLG1*, *CLN3*, *GSP1*, and *VHS1,* are involved in the cell cycle and/or maintenance of nucleolar organization and were also upregulated in co-culture with *kuratsuki Kocuria* (Figure 2, Table S2). Clg1 is a cyclin-like protein, which interacts with the cyclin-dependent kinase Pho85 [69,70]. Cln3 is activated in response to nutrient repletion induced by acetyl-CoA [71]. Gsp1 is a GTP-binding protein involved in the maintenance of nuclear organization [72]. Vhs1 suppresses cell cycle arrest during the G1-S transition [73].

Meanwhile, *TEM1*, which is involved in the termination of the M phase [74], was downregulated by less than 0.5-fold (Figure 3, Table S3); this is inconsistent with the upregulation of genes related to the meiosis/sporulation/cell cycle. In addition, the upregulation (Figure 2, Table S2) of *CLN3* is also inconsistent with the upregulation of meiosis-induced genes because G1 cyclin Cln3 blocks meiosis [75,76]. Therefore, it is uncertain whether sake yeast was about to undergo meiosis and enter the spore formation stage in co-culture with *kuratsuki Kocuria*. It is possible that various cells at different cell cycle stages existed.

### 3.6. Downregulation of Metabolic Genes in Co-Culture with kuratsuki Kocuria

The expression of other genes involved in metabolism was downregulated by less than 0.5-fold in the co-culture with *K. uropygioeca* TGY1127_2, except for *YIG1* (Figure 2, Table S2). Thus, among the 61 genes downregulated by less than 0.5-fold, fourteen genes, *ASN1*, *CHA1*, *COQ5*, *DAL3*, *ELO1*, *ENO2*, *FAA4*, *GPH1*, *GSH2*, *MET28*, *TDH1*, *TDH2*, *TDH3*, and *THI4*, were involved in metabolic regulation (Figure 3, Table S3). Asn1 encodes an asparagine synthetase [77]. Cha1 is an L-serine (L-threonine) deaminase [78]. Coq5 is a 2-hexaprenyl-6-methoxy-1,4-benzoquinone methyltransferase involved in ubiquinone biosynthesis [79]. Dal3 is a ureidoglycolate hydrolase [80]. Elo1 is a medium-chain acyl elongase [81]. Eno2 is a phosphopyruvate hydratase [82]. Faa4 is a long-chain fatty acyl-CoA synthetase [83]. Gph1 is a glycogen phosphorylase [84]. Gsh2 is a glutathione synthetase [56]. Met28 is a transcriptional activator that regulates sulfur metabolism [85]. Tdh1, Tdh2, and Tdh3 are the glyceraldehyde-3-phosphate dehydrogenases involved in glycolysis [86]. The glyceraldehyde-3-phosphate dehydrogenase activity generally increases with ethanol stress [87]. Thus, the downregulation of the glyceraldehyde-3-phosphate dehydrogenase genes was inconsistent with the increase in ethanol concentration during fermentation. Thi4 is a thiazole synthase [88]. These downregulated metabolic genes play important roles in growth processes, such as glycolysis. Thus, the cell activity of sake yeast decreased in the co-culture with *kuratsuki Kocuria*.

### 3.7. Expression Changes of Transporter Genes and Mitochondria-Related Genes in Co-Culture with the kuratsuki Kocuria

The hexose transporter gene *HXT11* [89] was 2.2-fold upregulated in the co-culture with *K. uropygioeca* TGY1127_2 (Figure 2, Table S2). Three transporter genes, *CTR3*, *OPT2*, and *PHO84*, were downregulated <0.5-fold (Figure 3, Table S3). Ctr3 is a copper transporter [90]. Opt2 is an oligopeptide transporter [91]. Pho84 is a phosphate transporter [92].

Although nine mitochondria-related genes, *CIT2*, *LSC1*, *MPC3*, *MRPL4*, *NCA3*, *NDE2*, *PET9*, *SFC1*, and *SLS1*, were upregulated more than 2-fold in co-culture with *kuratsuki Kocuria*, two genes (*SOM1* and *TIM50*) were downregulated less than 0.5-fold (Figures 2 and 3, Tables S2 and S3), thereby suggesting that *kuratsuki Kocuria* affected the mitochondrial system of sake yeast. Cit2 is a citrate synthase [93]. Lsc1 is a succinyl-CoA ligase [94]. Mpc3 is a mitochondrial pyruvate carrier [42]. Mrpl4 is a mitochondrial ribosomal protein YmL4 [95]. Nca3 regulates the mitochondrial expression of Fo-F1 ATP synthase subunits [96]. Nde2 is a mitochondrial NADH dehydrogenase [97]. Pet9 is an adenosine diphosphate/ATP translocator [98]. Sfc1 is a succinate–fumarate transporter [99]. Sls1 is a regulator of mitochondrial genes [100]. Som1 is a mitochondrial inner-membrane peptidase [101]. Tim50 is a translocase of the mitochondrial presequence translocase [102]. In addition, *PDC1* was also 2.3-fold upregulated in co-culture with *K. uropygioeca* TGY1127_2 (Figure 2, Table S2). Pdc1 is a key enzyme in alcoholic fermentation [103], strongly suggesting that ethanol fermentation is activated. However, this was inconsistent with the downregulation of the glycolytic enzyme genes *ENO2*, *TDH1*, *TDH2*, and *TDH3*. Further studies are needed to elucidate the state of sake yeast mitochondria in co-cultures with *kuratsuki Kocuria*.

### 3.8. Comparison of Gene Expression Changes in Co-Culture with the kuratsuki Kocuria and with Lactic Acid Bacteria

Mendes et al. [104] identified eleven upregulated (*CCC2*, *CIN5*, *CTR1*, *CTR3*, *EEB1*, *FIT1*, *FMP23*, *HMX1*, *SIT1*, *STL1*, and *TIS11*) and six downregulated (*ADH2*, *ATO3*, *CAR2*, *PHM8*, *SUL1*, and *XBP1*) genes in a co-culture with *Lactobacillus delbrueckii* subsp. *bulgaricus*. Ccc2 is a $Cu^{2+}$ transporting ATPase [105]. Cin5 is a transcription factor involved in the response to various stressors [106,107]. Ctr1 and Ctr3 are a copper transporter [90,108]. Interestingly, *CTR3* was 4.6-fold upregulated in response to co-culture with *L. delbrueckii* subsp. *bulgaricus* [104]. Thus, the expression of this gene is the opposite in co-cultures with *kuratsuki Kocuria* and lactic acid bacteria. Eeb1 is an alcohol *O*-octanoyltransferase [109]. Fit1 is a cell-wall mannoprotein [110]. Fmp23 is a putative protein involved in copper and iron metabolism [111]. Hmx1 is a heme oxygenase [112]. Sit1 is a ferrioxamine B permease [113]. Stl1 is a glycerol proton symporter [114]. Tis11 is an RNA-binding protein [115]. Adh2 is alcohol dehydrogenase [116]. Ato3 is a putative outward ammonium transporter [117]. Car2 is an ornithine transaminase [118]. Phm8 is a magnesium-dependent lysophosphatidic acid phosphatase [119]. Sul1 is a sulfate transporter [120], while Xbp1 is a stress-induced transcription factor [121]. Genes other than *CTR3* are not listed in Tables S2 and S3. Therefore, gene expression changes in *S. cerevisiae* differed completely in response to co-culture with *K. uropygioeca* TGY1127_2 and *L. delbruechii* subsp. *bulgaricus*.

Liu et al. [122] showed fifteen upregulated (*CIN5*, *DAK2*, *GPX2*, *HCH1*, *HSP104*, *PAU3*, *PAU19*, *PAU21*, *PAU22*, *PDR15*, *PHM8*, *RSB1*, *SRX1*, *TPS2*, and *TRR1*) and 11 downregulated (*BCY1*, *BUD25*, *HSC82*, *HTA2*, *HYR1*, *MXR2*, *OXR1*, *SCO1*, *TEM1*, *THI4*, and *TRX1*) genes in a co-culture with *Lactiplantibacillus plantarum*. Cin5 is a transcription factor involved in response to various stressors [106,107]. Dak2 is a dihydroxyacetone kinase [123]. Gpx2 is a glutathione peroxidase [124]. Hch1 is a chaperone-binding ATPase activator [125]. Hsp104 is involved in the response to multiple stressors [126]. Pau3, Pau19, Pau21, and Pau22 belong to the seripauperin family [127]. Pdr15 is a general stress response factor [128]. Phm8 is a magnesium-dependent lysophosphatidic acid phosphatase [119]. *PHM8* was downregulated and upregulated in co-cultures with *Lactobacillus delbrueckii* subsp. *bulgaricus* and *Lactiplantibacillus plantarum*, respectively. Rsb1 is involved in the sphingoid long-chain base release [129]. Srx1 is involved in the oxidative stress response [47]. Tps2

is trehalose-6-phosphate phosphatase [130]. Trr1 is a thioredoxin reductase [131]. Bcy1 is a cyclic adenosine-monophosphate-dependent protein kinase [132]. Bud25 is involved in bud-site selection [133]. Hsc82 is a chaperone and a paralog of Hsp82 [134]. Hta2 is a histone H2A [135]. *HTA2* was 0.56-fold downregulated in the co-culture with *kuratsuki Kocuria* (Table 1). Hyr1 is a glutathione peroxidase [124]. Mxr2 encodes a methionine sulphoxide reductase [136]. Oxr1 is involved in regulating V-ATPase [137]. Sco1 is a copper-binding protein involved in the function of cytochrome c oxidase [138]. Tem1 is a GTP-binding protein involved in the termination of M phase [74]. Thi4 is a thiazole synthase [88]. Trx1 is a thioredoxin [139].

The two upregulated genes, *PAU19* and *SRX1*, and the two downregulated genes, *TEM1* and *THI4*, are listed in Tables S2 and S3, respectively. As previously mentioned, *PAU19* and *SRX1* are stress-inducing genes. *TEM1* and *THI4* are involved in the termination of the M phase and thiamine biosynthesis, respectively. The other upregulated and downregulated genes differed between co-cultures with *K. uropygioeca* TGY1127_2 and *Lactiplantibacillus plantarum*.

Although no genes with similar expression changes were found between co-cultures with *K. uropygioeca* TGY1127_2 and *Lactobacillus delbrueckii* subsp. *bulgaricus*, four genes were found between co-cultures with *K. uropygioeca* TGY1127_2 and *Lactiplantibacillus plantarum*. In addition, although both *L. delbruechii* subsp. *bulgaricus* and *Lactiplantibacillus plantarum* are lactic acid bacteria, a similar expression pattern was observed only for *CIN5*. These results indicate that *S. cerevisiae* has different gene expression responses in the co-culture of different bacterial species.

## 4. Conclusions

We showed that *K. uropygioeca* TGY1127_2 interacted with sake yeast and altered its gene expression patterns. The genes involved in cell cycle/sporulation trended to be upregulated and those involved in metabolism trended to be downregulated in response to co-culture with *K. uropygioeca* TGY1127_2. The gene expression of yeast in co-culture with *kuratsuki Kocuria* differed from that in co-culture with lactic acid bacteria. This strongly suggests that *S. cerevisiae* exhibits different gene expression patterns in response to different bacteria during sake production. Thus, a novel combination of sake yeast and *kuratsuki bacteria* can lead to a unique sake that has never existed before.

Although our findings showed that the sake yeast changed its gene expression under co-existing with the *kuratsuki Kocuria*, these results cannot be directly applied to the relationship between sake yeast and *kuratsuki* bacteria in sake making. This is because the culture conditions are different. It is already known that the effect of *kuratsuki* bacteria differs by changing the type of *koji*, etc. In the future, it will be necessary to investigate how sake yeast is affected by *kuratsuki* bacteria in the sake-making environment.

**Supplementary Materials:** The following supporting information can be downloaded at: https://www.mdpi.com/article/10.3390/fermentation10050249/s1, Table S1: brix of culture. Table S2: more than 2-fold upregulated genes of sake yeast in co-culture with the *kuratsuki Kocuria*. Table S3: less than 0.5-fold downregulated genes of sake yeast in co-culture with the *kuratsuki Kocuria*. Figure S1: volcano plot between hold change and *p* value.

**Author Contributions:** Conceptualization, H.N.; methodology, H.N.; validation, H.N.; formal analysis, K.K.; investigation, K.K. and H.N.; writing, H.N.; visualization, H.N. All authors have read and agreed to the published version of the manuscript.

**Funding:** This work was supported by JSPS KAKENHI Grant Number 21H02109 (to H.N.).

**Institutional Review Board Statement:** Not applicable.

**Informed Consent Statement:** Not applicable.

**Data Availability Statement:** The original contributions presented in the study are included in the article and Supplementary Material, further inquiries can be directed to the corresponding author.

**Conflicts of Interest:** The authors declare no conflicts of interest.

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
