# Peer review of "Transcriptome Analysis of Sake Yeast in Co-Culture with kuratsuki Kocuria"

_fermentation, doi:10.3390/fermentation10050249_

Round 1

Reviewer 1 Report

Comments and Suggestions for Authors

The manuscript of Kobayashi and Nishida describes results on differential RNA expression when comparing the growth of S. cerevisiae and S. cerevisiae/Kocuria uropygiocea in a complex synthetic medium. The manuscript is exciting but needs improvements.

1. It needs to be emphasized that these experiments were performed in a synthetic medium, also in the title. The results may be very different in a natural medium for sake production.

2. Kocuria is one bacteria type in kuratsuki. Please add a paragraph about the kuratsuki bacteria in the introduction and why you selected Kocuria. 

3. I suggest using the text with the name of Kozuria species without kuratsuki.

4. Method: how did you remove the DNA? Digestion of DNA is one of the steps in RNA-seq.

5. Tables 1 and 2: Putting them in the supplement is better. For the main text, present these data in the graph.

6. When listing up-and-down-regulated genes, a context is missing. You have to give some possible roles of these regulated genes.

7. L29: phylogenetic lineages is not a proper word in this sentence.

8. L64: give some more explanation on Brix. What exactly does it represent, and how is it correlated with mg/L?

9. Why was the Kolmogorov-Smirnov test used? Why are more standard statistics not appropriate?

10. Why did you use only the read that mapped to a single locus?

11. Please add the acc. no. of the genome to which you mapped the reads.

12. L107: explain RPKM

13. L287-288: Where is this discussed in the text? To which paper does this correspond? 

Author Response

Thank you so much for your comments.

The manuscript of Kobayashi and Nishida describes results on differential RNA expression when comparing the growth of S. cerevisiae and S. cerevisiae/Kocuria uropygiocea in a complex synthetic medium. The manuscript is exciting but needs improvements.

Q1. It needs to be emphasized that these experiments were performed in a synthetic medium, also in the title. The results may be very different in a natural medium for sake production.

A1. We added the following in Materials and Methods, “Thus, monoculture and co-culture were performed in the artificial medium, which differs from the fermentation environment during sake making.”

Q2. Kocuria is one bacteria type in kuratsuki. Please add a paragraph about the kuratsuki bacteria in the introduction and why you selected Kocuria.

A1. We added the following in Introduction, “We isolated and identified two different strains belonging to the genus Kocuria, which were K. koreensis strain TGY1120_3 and K. uropygioeca strain TGY1127_2 [15]. We have used these two strains as representative kuratsuki bacteria in our experiments, especially K. uropygioeca strain TGY1127_2 [18][19][20].”

Q3. I suggest using the text with the name of Kozuria species without kuratsuki.

A3. We changed 17 “kuratsuki Kocuria” to “K. uropygioeca TGY1127_2”.

Q4. Method: how did you remove the DNA? Digestion of DNA is one of the steps in RNA-seq.

A4. We added the following in Materials and Methods, “DNA was digested and”.

Q5. Tables 1 and 2: Putting them in the supplement is better. For the main text, present these data in the graph.

A5. We transferred Tables 1 and 2 into Tables S1 and S2 as supplement. We added Figs. 2 and 3.

Q6. When listing up-and-down-regulated genes, a context is missing. You have to give some possible roles of these regulated genes.

A6. In Tables S1 and S2, the product shows the gene function. Some products are not popular. Thus, we added the references in these tables with additional eight references [140]-[147]. In Figs. 2 and 3, color coding was done based on gene function.

Q7. L29: phylogenetic lineages is not a proper word in this sentence.

A7. We omitted the following, “Beer, sake, and wine yeasts form different phylogenetic lineages.”

Q8. L64: give some more explanation on Brix. What exactly does it represent, and how is it correlated with mg/L?

A8. We added the following in Materials and Methods, “Brix is a measure of the percentage of soluble solids in syrups and the products. Each degree of Brix is equal to 1% concentration of sugar in solution when measured at 20 °C (foodscience-avenue.com).”

Q9. Why was the Kolmogorov-Smirnov test used? Why are more standard statistics not appropriate?

A9. We added the following in Materials and Methods, “The Kolmogorov-Smirnov test was performed for comparing the two samples using R software (The R Project for Statistical Computing, http://www.R-projet.org/ (accessed on 25 January 2024)). The Kolmogorov-Smirnov test is sensitive to differences in both location and shape of the empirical cumulative distribution functions of the two samples, which is one of the most useful nonparametric methods for comparison of two samples.”

Q10. Why did you use only the read that mapped to a single locus?

A10. We added the following in Results and discussion, “because their origins were unclear”.

Q11. Please add the acc. no. of the genome to which you mapped the reads.

A11. We added the following in Materials and Method, “(NCBI RefSeq ID:”.

Q12. L107: explain RPKM

A12. We added the following, “(the ratio of reads per kilobase of exon per million mapped reads)”.

Q13. L287-288: Where is this discussed in the text? To which paper does this correspond?

A13. We omitted the following, “Sake yeasts produce several esters and organic acids outside their cells during sake production, which determines the flavor and taste of sake.”

Reviewer 2 Report

Comments and Suggestions for Authors

The authors of the manuscript "fermentation-2983734" report on the gene expression changes in sake yeast (Saccharomyces cerevisiae) when co-cultured with Kuratsuki Kocuria bacteria through transcriptome analysis. This study reveals the impact of Kuratsuki bacteria on the gene expression of sake yeast, which aids in understanding the microbial basis for the unique flavor formation in sake and may provide new strategies for improving the sake production process through microbial interactions. However, some details in this manuscript need to be noticed.

1. In the abstract, the phrase "Fourteen and six stress-induced genes were upregulated and downregulated, respectively," could be clarified to read: "Among the stress-induced genes, fourteen were upregulated, and six were downregulated."

2. In the introduction section, the statements "Beer, sake, and wine yeasts form different phylogenetic lineages," and "The effects of co-culture of S. cerevisiae with bacteria were reported not only in the beverage/food fermentation process but also in industrial ethanol production process," are somewhat redundant and not closely related to the content of the research. These sentences could be omitted.

3. The introduction is somewhat brief and could benefit from a more detailed discussion on the current state of research regarding mixed-strain fermentation with sake yeast and the current state of research on Kuratsuki bacteria in sake fermentation.

4. The introduction does not explain the reasons for studying the diversity of Kuratsuki bacteria. It would be beneficial to include references and experimental justification for this aspect.

5. In the introduction, lines 40-47, there is a lack of clear logical connection between the first part, which discusses the diversity of Kuratsuki bacteria, and the subsequent statement that "Kuratsuki bacteria may interact with sake yeast to change the metabolism as well as lactic acid bacteria." The writing order should be adjusted to improve coherence.

6. The materials and methods section should be supplemented with the statistical analysis method used.

7. In Figure 1, it is necessary to include a caption explaining what the "triangles" and "circles" represent for each experimental group, and error bars should be added to the figure.

8. In the Results and Discussion section, it would be appropriate to select some of the more significant genes and discuss and explain the possible reasons for their upregulation or downregulation in conjunction with relevant literature (For example:NADH accumulation in S. cerevisiae is known to stimulate TDH1 expression (Valadi et al. 2004) which, in the context of the current study, suggests that ethanol-stressed cells experience reductive stress. Moreover, Some of its fragments have been described as antimicrobial peptides against a variety of yeasts(Branco et al., 2014),this also suggests that the upregulation of these genes could also be due to the sake yeast's action of killer activity against other yeasts).

Author Response

Thank you so much for your comments.

The authors of the manuscript "fermentation-2983734" report on the gene expression changes in sake yeast (Saccharomyces cerevisiae) when co-cultured with Kuratsuki Kocuria bacteria through transcriptome analysis. This study reveals the impact of Kuratsuki bacteria on the gene expression of sake yeast, which aids in understanding the microbial basis for the unique flavor formation in sake and may provide new strategies for improving the sake production process through microbial interactions. However, some details in this manuscript need to be noticed.

Q1. In the abstract, the phrase "Fourteen and six stress-induced genes were upregulated and downregulated, respectively," could be clarified to read: "Among the stress-induced genes, fourteen were upregulated, and six were downregulated."

A1. According to your comment, we revised it.

Q2. In the introduction section, the statements "Beer, sake, and wine yeasts form different phylogenetic lineages," and "The effects of co-culture of S. cerevisiae with bacteria were reported not only in the beverage/food fermentation process but also in industrial ethanol production process," are somewhat redundant and not closely related to the content of the research. These sentences could be omitted.

A2. We omitted the following, “Beer, sake, and wine yeasts form different phylogenetic lineages” and “not only in the beverage/food fermentation process but also”.

Q3. The introduction is somewhat brief and could benefit from a more detailed discussion on the current state of research regarding mixed-strain fermentation with sake yeast and the current state of research on Kuratsuki bacteria in sake fermentation.

A3. We added the following, “We isolated and identified two different strains belonging to the genus Kocuria, which were K. koreensis strain TGY1120_3 and K. uropygioeca strain TGY1127_2 [15]. We have used these two strains as representative kuratsuki bacteria in our experiments, especially K. uropygioeca strain TGY1127_2 [18][19][20].” Please see A4 and A5.

Q4. The introduction does not explain the reasons for studying the diversity of Kuratsuki bacteria. It would be beneficial to include references and experimental justification for this aspect.

A4. We added the following, “Although research on kuratsuki yeasts have been performed, research on kuratsuki bacteria has been considered unnecessary. As a result, there was no information on difference of kuratsuki bacteria among different sake breweries.” Please see A3 and A5.

Q5. In the introduction, lines 40-47, there is a lack of clear logical connection between the first part, which discusses the diversity of Kuratsuki bacteria, and the subsequent statement that "Kuratsuki bacteria may interact with sake yeast to change the metabolism as well as lactic acid bacteria." The writing order should be adjusted to improve coherence.

A5. We added the following, “We used the taste recognition device to confirm whether the co-culture would change the taste. Different effects of kuratsuki bacteria were found on different types of koji and different strains of sake yeast [13][18][19].” Please see A3 and A4.

Q6. The materials and methods section should be supplemented with the statistical analysis method used.

A6. We added the following, “The Kolmogorov-Smirnov test was performed for comparing the two samples using R software (The R Project for Statistical Computing, http://www.R-projet.org/ (accessed on 25 January 2024)). The Kolmogorov-Smirnov test is sensitive to differences in both location and shape of the empirical cumulative distribution functions of the two samples, which is one of the most useful nonparametric methods for comparison of two samples.”

Q7. In Figure 1, it is necessary to include a caption explaining what the "triangles" and "circles" represent for each experimental group, and error bars should be added to the figure.

A7. We revised Figure 1.

Q8. In the Results and Discussion section, it would be appropriate to select some of the more significant genes and discuss and explain the possible reasons for their upregulation or downregulation in conjunction with relevant literature (For example:NADH accumulation in S. cerevisiae is known to stimulate TDH1 expression (Valadi et al. 2004) which, in the context of the current study, suggests that ethanol-stressed cells experience reductive stress. Moreover, Some of its fragments have been described as antimicrobial peptides against a variety of yeasts(Branco et al., 2014),this also suggests that the upregulation of these genes could also be due to the sake yeast's action of killer activity against other yeasts).

A8. We described TDH1 in the following, “Tdh1, Tdh2, and Tdh3 are the glyceraldehyde-3-phosphate dehydrogenases involved in glycolysis [86]. The glyceraldehyde-3-phosphate dehydrogenase activity generally increases with ethanol stress [87]. Thus, the downregulation of the glyceraldehyde-3-phosphate dehydrogenase genes was inconsistent with the increase in ethanol concentration during fermentation.” and “Pdc1 is a key enzyme in alcoholic fermentation [103], strongly suggesting that ethanol fermentation is activated. However, this was inconsistent with the downregulation of the glycolytic enzyme genes ENO2, TDH1, TDH2, and TDH3. Further studies are needed to elucidate the state of sake yeast mitochondria in co-cultures with kuratsuki Kocuria.” We cannot describe antimicrobial peptides in this paper.

We transferred Tables 1 and 2 into Tables S1 and S2 as supplement. We added Figs. 2 and 3. In Figs. 2 and 3, color coding was done based on gene function. In Tables S1 and S2, the product shows the gene function. Some products are not popular. Thus, we added the references in these tables with additional eight references [140]-[147]. In addition, we added the following in Conclusion, “The genes involved in cell cycle/sporulation trended to be upregulated and those involved in metabolism trended to be downregulated in response to co-culture with K. uropygioeca TGY1127_2.”

Round 2

Reviewer 1 Report

Comments and Suggestions for Authors

The manuscript has been corrected accordingly.

Author Response

Thank you so much for your reviewing.

Reviewer 2 Report

Comments and Suggestions for Authors

I am delighted to receive your response and the revised version of the manuscript. However, there are still some sections of the manuscript that require further revision.

Logical Issues in the Introduction Section:

The concern raised in point Q5 of my previous review has not been adequately addressed. The fact that you have isolated different strains of Kuratsuki from sake breweries, as described in your manuscript, does not sufficiently demonstrate the existence of an interaction between Kuratsuki and sake yeast.

Regarding Figure 1 and Error Bars:

Necessity of Error Bars: We have observed that despite the previous suggestions to add error bars, the revised Figure 1 does not appear to incorporate this. While we understand that the Kolmogorov-Smirnov test does not directly produce error bars, if Figure 1 is intended to present experimental data points, the inclusion of error bars is essential for demonstrating the variability of the data. Please consider adding error bars to Figure 1 to reflect the repeatability and reliability of the experimental measurements.

Completeness of Figure Caption: We recommend that the authors provide a detailed figure caption explaining the symbols used in the figure (such as "triangles" and "circles") and what specific experimental conditions they represent. Additionally, if error bars have been added, the caption should include an explanation of what the error bars represent, such as the standard deviation (SD), standard error (SEM), or another statistical measure.

Regarding Figures 2 and 3:

You may consider replacing Figures 2 and 3 with a volcano plot.

Discussion Section:

Expand on the implications of the findings, particularly how they relate to and advance the current understanding of the interactions between sake yeast and Kuratsuki. Address any limitations encountered in the study and how these might influence the interpretation of the results. Explore the potential applications of the research, including how the insights gained could be leveraged to improve sake production processes. Discuss whether the results align with or diverge from previous studies and, if so, what new insights are provided by the current work.

Author Response

Reply to Reviewer #2:

Thank you so much for your reviewing.

Q1. The concern raised in point Q5 of my previous review has not been adequately addressed. The fact that you have isolated different strains of Kuratsuki from sake breweries, as described in your manuscript, does not sufficiently demonstrate the existence of an interaction between Kuratsuki and sake yeast.

A1. We added the following, “Thus, the taste of sake with kuratsuki bacteria differed from that without kuratsuki bacteria. We are performing research focusing on the microbial interaction.”

Q2. Necessity of Error Bars: We have observed that despite the previous suggestions to add error bars, the revised Figure 1 does not appear to incorporate this. While we understand that the Kolmogorov-Smirnov test does not directly produce error bars, if Figure 1 is intended to present experimental data points, the inclusion of error bars is essential for demonstrating the variability of the data. Please consider adding error bars to Figure 1 to reflect the repeatability and reliability of the experimental measurements.

A2. We used NOT mean but median as representative value of three measured values because median has higher robustness than mean. Error bars should be added to mean (not median). Boxplot was useful for median. But, three values are small to show boxplot. So, we revised Fig. 1 with three measured values with caption. HOWEVER, the editorial staff took a mistake not to replace the figure. I cannot understand why you cannot check the revised figure (original). It was a problem that some points were identical. So, the three points were not seen. Thus, we added supplement table (Table S1).

Q3. Completeness of Figure Caption: We recommend that the authors provide a detailed figure caption explaining the symbols used in the figure (such as "triangles" and "circles") and what specific experimental conditions they represent. Additionally, if error bars have been added, the caption should include an explanation of what the error bars represent, such as the standard deviation (SD), standard error (SEM), or another statistical measure.

A3. Please see A2.

Q4. You may consider replacing Figures 2 and 3 with a volcano plot.

A4. According to the comments of Reviewer #1, we transferred Tables to Suppl. Tables and added figures. So, the member of genes is identical to suppl. Tables. Thus, we added supplement figure of volcano plot (Fig. S1).

Q5. Expand on the implications of the findings, particularly how they relate to and advance the current understanding of the interactions between sake yeast and Kuratsuki. Address any limitations encountered in the study and how these might influence the interpretation of the results. Explore the potential applications of the research, including how the insights gained could be leveraged to improve sake production processes. Discuss whether the results align with or diverge from previous studies and, if so, what new insights are provided by the current work.

A5. We added the following, “Although our findings showed that the sake yeast changed its gene expression under co-existing the kuratsuki Kocuria, these results cannot be directly applied to the relationship between sake yeast and kuratsuki bacteria in sake making. This is because the culture conditions are different. It is already known that the effect of kuratsuki bacteria differs by changing the type of koji, etc. In the future, it will be necessary to investigate how sake yeast is affected by kuratsuki bacteria in the sake making environment.”